# Generalized hydrodynamics of integrable quantum circuits

**Friedrich Hübner[1], Eric Vernier[2] and Lorenzo Piroli[3]**

**1** Department of Mathematics, King's College London, Strand, London WC2R 2LS, U.K.
**2** Laboratoire de Probabilités, Statistique et Modélisation CNRS,
Université Paris Cité, Sorbonne Université Paris, France
**3** Dipartimento di Fisica e Astronomia, Università di Bologna and INFN,
Sezione di Bologna, via Irnerio 46, I-40126 Bologna, Italy

## Abstract

Quantum circuits make it possible to simulate the continuous-time dynamics of a many-body Hamiltonian by implementing discrete Trotter steps of duration $\tau$. However, when $\tau$ is sufficiently large, the discrete dynamics exhibit qualitative differences compared to the original evolution, potentially displaying novel features and many-body effects. We study an interesting example of this phenomenon, by considering the integrable Trotterization of a prototypical integrable model, the XXZ Heisenberg spin chain. We focus on the well-known bipartition protocol, where two halves of a large system are prepared in different macrostates and suddenly joined together, yielding non-trivial nonequilibrium dynamics. Building upon recent results and adapting the generalized hydrodynamics (GHD) of integrable models, we develop an exact large-scale description of an explicit one-dimensional quantum-circuit setting, where the input left and right qubits are initialized in two distinct product states. We explore the phenomenology predicted by the GHD equations, which depend on the Trotter step and the gate parameters. In some phases of the parameter space, we show that the quantum-circuit large-scale dynamics is qualitatively different compared to the continuous-time evolution. In particular, we find that a single microscopic defect at the junction, such as the addition of a single qubit, may change the nonequilibrium macrostate appearing at late time.

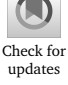
doi:10.21468/SciPostPhys.18.4.135

## Contents



# 1 Introduction

The Trotter-Suzuki decomposition [1,2] is at the basis of the idea of quantum simulation [3]. Up to a controlled error, this decomposition allows us to break down the continuous-time evolution of a many-body system into a series of elementary two-body interactions of duration $\tau$, which could be implemented as *quantum gates* in a quantum circuit [4]. The Trotter time $\tau$ controls the accuracy and determines the number of gates applied. Unfortunately, external noise limits the number of operations which can be carried out in the noisy-intermediate-scale quantum (NISQ) devices [5] available today. Therefore, since simulating the continuous-time dynamics typically requires application of many quantum gates, full-fledged quantum simulation remains a non-trivial challenge [6].

Motivated by this picture, the past few years have witnessed growing interest in genuinely discrete dynamics where the Trotter time $\tau$ is not small [7,8]. The reason behind this interest is two-fold. On the one hand, these models may be easier to implement in current quantum platforms [9–13], since a non-trivial evolution may be observed over a small number of Trotter steps. On the other hand, discrete dynamics may exhibit qualitative differences compared to the original Hamiltonian system, with novel phenomena and many-body effects.

For typical Hamiltonians, recent work [14] has shown that increasing indefinitely the Trotter time $\tau$ leads to a sharp transition, see also [15–23]: when $\tau$ is increased beyond a threshold value, approximation errors become uncontrolled at large times. As a consequence, the evolution becomes chaotic, and one expects the rapid onset of local-equilibrium infinite-temperature physics [24, 25]. This mechanism, however, is not ubiquitous and a different behavior is expected, for instance, in the presence of integrability, where extensively many local conservation laws prevents the onset of quantum chaos [26–28].
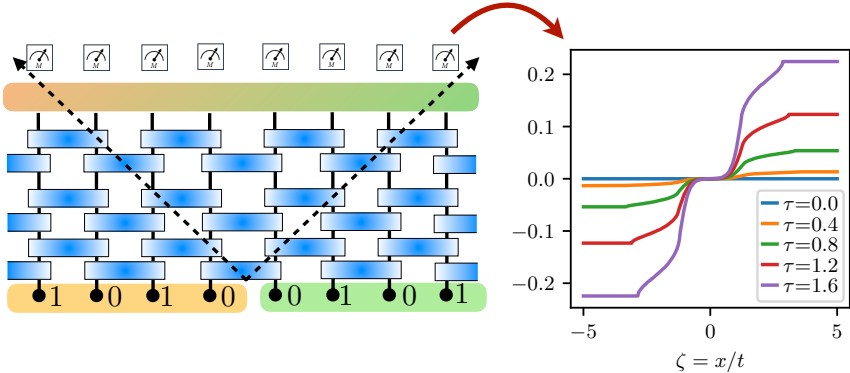

Figure 1: Sketch of the bipartition protocol. The system is initialized by joining at the origin two product states (the figure corresponds to the Néel and anti-Néel states) and evolved by applying a homogeneous quantum circuit, Trotterizing the integrable XXZ continuous-time dynamics ($\tau$ is the duration of the Trotter step). A lightcone spreads ballistically from the junction, inside of which non-trivial dynamics takes place. We study the profiles of local observables arising at late times. The plot shows the example of the staggered spin-$z$ magnetization and is obtained by solving the GHD equations, cf. Sec. 4.

Integrable quantum circuits providing a Trotterization of integrable Hamiltonians were introduced in Refs. [29–31]. In the past few years they have received significant attention due to their dynamical features [32–35], their rich underlying mathematical structures [36–39] and seminal work demonstrating experimental implementation in current quantum devices [40–42]. Prominently, Ref. [33] has shown the existence of Trotter transitions in integrable quantum circuits, manifesting in the correlations of the so-called generalized Gibbs ensemble (GGE) [26–28]. As GGEs describe the late-time physics of integrable systems, this result is interesting, as it implies that the information on the discretized structure of the dynamics is not washed out at coarse-grained time scales.

In this work, we build upon the results of Ref. [33] (which focused on homogeneous, translation-invariant settings) and investigate the physics of integrable quantum circuits in more general non-equilibrium situations. We focus in particular on the prototypical model of the XXZ Heisenberg spin chain [43] and consider the simplest case where the system is prepared in an inhomogeneous state, namely the so-called bipartition protocol [44–47]. The latter consists in a special type of *quantum quench* [48], where the two halves of a 1D system are prepared in distinct macrostates, and subsequently left to evolve according to the system unitary dynamics, cf. Fig. 1. While conceptually simple, this problem is hard. For integrable systems, its understanding has required significant theoretical work over the past decade, culminating in the development of the so-called generalized hydrodynamics (GHD) [49, 50], a very versatile and powerful theory.

In the bipartition protocol, the GHD predicts that local subsystems at a given distance $x$ from the junction equilibrate to local stationary states, depending only on the ratio $x/t$, defining a reference frame moving away from the origin. Such stationary states are characterized by space-dependent GGEs, which are quantitatively determined by a solution to the hydrodynamic equations [47]. The latter depend on the features of the specific model considered, including its particle content and symmetries. For this reason, while GHD provides a very general framework, the qualitative features of the dynamics can vary greatly from model to model, leading to a series of works aimed at exploring the phenomenology of the biparition protocol in different integrable systems [49–64].

The goal of this work is to apply the GHD framework to study the bipartition protocol in the XXZ integrable quantum circuit. Based on the known spectral properties of the model, we first derive the GHD equations and study their dependence on the Trotter step and circuit parameters. We consider an explicit protocol where the input left and right qubits are initialized in two distinct product states and evolved by applying a homogeneous quantum circuit. We find that the qualitative features of the discrete dynamics may vary greatly from the continuous-time evolution. Most prominently, we show that a single microscopic defect at the junction, such as the addition of a single qubit, may change the nonequilibrium macrostate appearing at late time. Our work complements Ref. [33], further exploring the interesting dynamical features of integrable quantum circuits in many-body settings.

The rest of this work is organized as follows. We begin in Sec. 2 introducing the XXZ quantum circuit and its Bethe Ansatz solution. In Sec. 3 we discuss the bipartition protocol and derive the necessary technical ingredients for its analytic description. They include obtaining the GGEs corresponding to different initial states, generalizing the results of Ref. [33], and deriving the GHD equations for integrable quantum circuits. Our results are discussed in detail in Sec. 4, where we provide predictions for the system dynamics at different points in the parameter space, by numerically solving the GHD equations. Our conclusions are presented in Sec. 5, while the appendices contain the most technical aspects of our work.

## 2 The model

### 2.1 The integrable quantum circuit

We consider a system of $L$ qubits arranged in a one-dimensional (1$D$) array. We denote by $|0\rangle_j$, $|1\rangle_j$ the basis states associated with the local Hilbert space at position $j$. The dynamics consists in the repeated (Floquet) application of the unitary operator

$$U(\tau) = U_e(\tau)U_o(\tau), \tag{1}$$

where

$$U_o(\tau) = \prod_{n=1}^{L/2} V_{2n,2n+1}(\tau), \tag{2}$$

$$U_e(\tau) = \prod_{n=1}^{L/2} V_{2n-1,2n}(\tau), \tag{3}$$

and

$$V_{n,n+1}(\tau) = e^{-i\frac{\tau}{4}\left[\sigma_n^x \sigma_{n+1}^x + \sigma_n^y \sigma_{n+1}^y + \Delta(\sigma_n^z \sigma_{n+1}^z - \mathbb{1})\right]}, \tag{4}$$

with $\tau \in \mathbb{R}$, and where periodic boundary conditions are assumed. $U(\tau)$ defines a brickwork quantum circuit which is a discrete version of the continuous-time evolution driven by the XXZ Heisenberg Hamiltonian [29, 65]

$$H = \frac{1}{4}\sum_{j=1}^{L}\left[\sigma_j^x \sigma_{j+1}^x + \sigma_j^y \sigma_{j+1}^y + \Delta(\sigma_j^z \sigma_{j+1}^z - \mathbb{1})\right], \tag{5}$$

where $\sigma_j^\alpha$ are the Pauli matrices acting at position $j$, with $\sigma_{L+1}^\alpha = \sigma_1^\alpha$. Indeed, we have

$$e^{-iHt} = \lim_{M\to\infty} U(t/M)^M. \tag{6}$$

In other words, for fixed time $t$, the continuous time evolution is recovered by taking $\tau \to 0$ and the circuit depth $M \to \infty$, keeping the product $\tau M = t$ constant.

For a given Hamiltonian, the Suzuki-Trotter decomposition is not unique, but the above discretization preserves integrability [29,30]. Namely, the Floquet operator (1) features an extensive number of local and quasilocal conserved operators known as charges [29–31]. For small $\tau$, they can be thought of as a deformation of those of the Hamiltonian (5): for each charge $Q_k$, with $[Q_k, H] = 0$, we have two new operators $\tilde{Q}_k^{\pm}(\tau)$ with $[\tilde{Q}_k^{\pm}(\tau), U(\tau)] = 0$. As pointed out in Refs. [29,31], the charges $\tilde{Q}_k^{\pm}(\tau)$ break the single-site translation symmetry. Instead, the operation leaving the set of conserved charges invariant is the combination of the single-site translation and a change of sign of the staggering parameter $x$ introduced in the next section.

## 2.2 The Bethe Ansatz solution

The integrability of the model allows one to obtain the exact spectrum of the Floquet operator $U(\tau)$ [36–38], which we now review. It is first important to discuss the phase diagram of the model. To this end, we introduce the parameters [30]

$$\gamma = \arccos\left[\sin(\Delta\tau/2)/\sin(\tau/2)\right], \tag{7a}$$

$$x = i \operatorname{arcsinh}\left[\sin(\gamma)\tan(\tau/2)\right], \tag{7b}$$

and distinguish two cases. First, if $\gamma = i\eta$, with $\eta \in \mathbb{R}$, then $x \in \mathbb{R}$. In this case, we say that the model is in the gapped or massive regime. As it will be clear from the subsequent discussion, the name comes from the fact that the spectrum of $U(\tau)$ can be described in terms of quasiparticles with similar features as those of the XXZ Hamiltonian (5) in the gapped regime $|\Delta| > 1$. Second, if $\gamma \in \mathbb{R}$ then $x$ is purely imaginary. In this case, we say that the model is in the gapless or massless regime. The phase diagram as a function of $\Delta$ and $\tau$, which parametrize the gates (4), can be found in Ref. [33]. For any $\Delta$, there is an infinite sequence of transitions as we increase $\tau$. The first phase transition takes place at the value

$$\tau_{\text{th}}(\Delta) = \frac{2\pi}{\Delta + 1}. \tag{8}$$

For $\Delta > 1$ ($\Delta \leq 1$), the system is in a gapped (gapless) phase for $\tau \leq \tau_{\text{th}}(\Delta)$.

The spectrum of $U(\tau)$ is organized into sectors labeled by the number $M$ of stable quasiparticles. The latter are magnonic excitations and, accordingly, $M$ is the quantum number associated with the conserved operator $\hat{M} = \sum_j (\mathbb{1} - \sigma_j^z)/2$. In a given $M$-sector, the eigenstates are parametrized by sets of complex numbers $\{p_j\}_{j=1}^M$, satisfying the so-called Bethe equations [36]

$$\left[\frac{f_x^+(p_i)}{f_x^-(p_i)}\right]^{\frac{L}{2}} = \prod_{k \neq j}^M \frac{\sinh\left(p_j - p_k + i\gamma\right)}{\sinh\left(p_j - p_k - i\gamma\right)}, \tag{9}$$

where

$$f_x^{\pm}(p) = \sinh\left(p + i\frac{x}{2} \pm i\frac{\gamma}{2}\right)\sinh\left(p - i\frac{x}{2} \pm i\frac{\gamma}{2}\right). \tag{10}$$

Each number $p_j$ is associated to a stable quasiparticle and is related to the corresponding quasimomentum or rapidity, denoted by $\lambda_j$. The precise relation between the two depends on the phase of the model: in the gapless regime, we have $\lambda_j = p_j$, while in the gapped regime $\lambda_j = ip_j$. For finite $M$, it is non-trivial to find all the solutions of the Bethe equations. However, the picture greatly simplifies in the thermodynamic limit $L \to \infty$, where the spectrum can be described via the so-called Thermodynamics Bethe Ansatz (TBA) formalism [66]. While

initially developed for the study of integrable Hamiltonians, the TBA can be straightforwardly extended to the quantum-circuit setting [33].

In essence, the TBA formalism is based on the string hypothesis [66] which postulates how the solutions to the Bethe equations organize themselves in the complex plane in the limit $L \to \infty$, with the density $D = M/L$ kept fixed. In particular, the string hypothesis states that the rapidities organize themselves into sets of $n$ elements forming a "string" of length $n$, which is interpreted as a bound state of $n$ quasiparticles. Each string features a center, $\lambda$, which is the bound-state quasimomentum. Therefore, we can associate to each macrostate a set of functions $\rho_n(\lambda)$: in a large system of size $L$, $L\rho_n(\lambda)d\lambda$ yields the number of $n$-quasiparticle bound states with rapidities in the interval $[\lambda, \lambda + d\lambda]$. In order to achieve a complete thermodynamic description, the TBA formalism also introduces the set of hole distribution functions for the $n$-string centers, denoted by $\rho_n^h(\lambda)$. The hole distribution functions are analogous to the distribution of vacancies (i.e., unoccupied states) in the ideal Fermi gas at finite temperature.

The precise form of the rapidity and hole distribution functions depends once again on the phase of the model. We therefore separate the discussion between the two cases in the following. Before proceeding, it is important to mention that the validity of the string hypothesis has been verified in a number of cases, including the present quantum-circuit setting, where the resulting predictions have been tested numerically against independent tensor-network computations [33].

### 2.2.1 The TBA in the gapped regime

In the gapped regime, the string length can take any possible integer value, $n \geq 1$, while the string centers satisfy $\lambda \in [-\pi/2, \pi/2]$. By standard techniques [66], one can then take the thermodynamic limit of the Bethe equations (9), resulting in an infinite set of coupled integral equations for the distribution functions $\rho_n(\lambda)$, $\rho_n^h(\lambda)$. Namely, introducing

$$\rho_n^t(\lambda) = \rho_n(\lambda) + \rho_n^h(\lambda), \tag{11}$$

the thermodynamic limit of the Bethe equations read

$$\rho_n^t(\lambda) = a_n^{(x/2)}(\lambda) - \sum_{m=1}^{\infty} (a_{nm} * \rho_m)(\lambda), \tag{12}$$

where

$$(f * g)(\lambda) := \int_{-\pi/2}^{\pi/2} d\mu f(\mu - \lambda) g(\mu), \tag{13}$$

while we introduced the notation

$$f^{(x)}(\lambda) = (f(\lambda + x) + f(\lambda - x))/2, \tag{14}$$

and defined

$$a_{nm}(\lambda) = (1 - \delta_{nm}) a_{|n-m|}(\lambda) + 2a_{|n-m|+2}(\lambda) + \ldots + 2a_{n+m-2}(\lambda) + a_{n+m}(\lambda), \tag{15}$$

with

$$a_n(\lambda) = \frac{\sinh(n\eta)}{\pi[\cosh(n\eta) - \cos(2\lambda)]}. \tag{16}$$

Introducing the ratios

$$\eta_n(\lambda) = \rho_n^h(\lambda)/\rho_n(\lambda), \tag{17}$$

we can also rewrite

$$\rho_n(\lambda)[1 + \eta_n(\lambda)] = a_n^{(x/2)}(\lambda) - \sum_{m=1}^{\infty}(a_{nm} * \rho_m)(\lambda). \tag{18}$$

Note that for $x = 0$, the above equations recover the standard TBA description of the homogeneous XXZ Heisenberg chain [66]. Eq. (9) is valid for any eigenstate of the system, and the dependence in the eigenstate lies in the set of rapidities $\{p_j\}$. In order to derive Eq. (12) one follows the so-called string hypothesis [66], assuming that in the thermodynamic limit the $p$'s arrange into strings of length $n$, whose centers are distributed according to $\rho_n(\lambda)$. Eq. (9) then translates into Eq. (12), which relates $\rho_n(\lambda)$ to the total density $\rho_n^t(\lambda)$. Similar to Eq. (9), Eq. (12) holds for any macrostate, and the dependence in a given macrostate is given by an additional relation between $\rho_n(\lambda)$ and $\rho_n^t(\lambda)$ (in other words, fixing $\rho_n^t/\rho_n$ specifies a macrostate uniquely). While $\rho_n(\lambda)$ is sufficient to compute expectation values of conserved quantities, $\rho_n^t(\lambda)$ is needed to compute other quantities, for instance the entropy of the macrostate $\rho_n(\lambda)$.

As we will see in the next section, the GHD is based on the fact that quasiparticles move ballistically through the system with a given effective velocity $v_n^{\text{eff}}(\lambda)$. These velocities are determined the distribution functions $\rho_n(\lambda)$, $\rho_n^h(\lambda)$ and in particular are obtained as the solution to the system

$$\rho_n^t(\lambda) v_n^{\text{eff}}(\lambda) = \frac{1}{2\pi} E_n'(\lambda) - \sum_{m=1}^{\infty} \left(a_{nm} * (\rho_m v_m^{\text{eff}})\right)(\lambda), \tag{19}$$

where $E_n(\lambda)$ will be defined in the following section, cf. Eq. (62). As we will discuss, Eqs. (19) are derived by generalizing the definition given in the Hamiltonian setting in a straightforward way.

In principle, the rapidity distribution functions give us a complete description of the system in the thermodynamic limit. Yet, while simple expressions are known for the expectation value of the local conserved charges, it is generally hard to derive formulas for the correlation functions – a task which, in the XXZ Hamiltonian model, has been carried out only in a few special cases [67–70]. In this work, we will restrict ourselves to study three simple observables. First, we consider the local averaged magnetization

$$\overline{m} = \frac{\langle \sigma_j^z \rangle + \langle \sigma_{j+1}^z \rangle}{2}. \tag{20}$$

Exploiting the two-site shift-invariance of the model, $\overline{m}$ coincides with the normalized expectation value of the total spin-$z$ magnetization, $S^z = \sum_j \sigma_j^z$, which is conserved by the dynamics. In fact, $(1 - S^z)/2$ counts the number of quasi-particles in each eigenstate, so that we have the simple expression

$$\overline{m} = 1 - 2 \sum_{n=1}^{\infty} \int_{-\pi/2}^{\pi/2} d\lambda \, n \rho_n(\lambda). \tag{21}$$

The second observable that we consider is the local current associated with the local spin-$z$ operator. Its form can be inferred from the continuity equation

$$U^{\dagger}(\tau)\sigma_j^z U(\tau) - \sigma_j^z = J_{j+1} - J_j, \tag{22}$$

from which we identify

$$J_{2j} = \mathcal{N}_x^{-1} \left[ 2(\sigma_{2j-1}^+ \sigma_{2j}^- - \sigma_{2j-1}^- \sigma_{2j}^+) + i \sinh(ix)(\sigma_{2j-1}^z - \sigma_{2j}^z) \right], \tag{23}$$

where

$$\mathcal{N}_x = \frac{i(1 + \cosh 2ix)}{2 \sinh ix}. \tag{24}$$

Similarly, the expression for $J_{2j+1}$ is obtained from (23) by shifting all sites by one and changing $x$ to $-x$, or, equivalently, by simply making the change $2j-1 \rightarrow 2j+1$. The expectation value of the current associated to a conserved charge is known by recent Bethe Ansatz results [71–73] and reads

$$\langle J_{2j} \rangle = -2 \sum_{n=1}^{\infty} \int_{-\pi/2}^{\pi/2} d\lambda \, n \rho_n(\lambda) v_n^{\text{eff}}(\lambda). \tag{25}$$

Finally, we consider the one-point function $\langle \sigma_j^z \rangle$. Since the model breaks single-site translation-symmetry, this expectation value can not be reduced to the value of a conserved charge. In fact, this is a non-trivial one-point function, for which Ref. [33] derived the formula (expressed in the notation of this paper)

$$\langle \sigma_j^z \rangle = 1 - 2 \sum_{n=1}^{\infty} n \int_{-\pi/2}^{\pi/2} d\lambda \, \theta_n(\lambda) b_n^{\text{dr}}(\lambda), \tag{26}$$

where we defined $b_n^{\text{dr}}(\lambda)$ as the solution to the equations

$$b_n^{\text{dr}}(\lambda) = b_n(\lambda) - \sum_m [a_{nm} * (\theta_n(\lambda) b_m^{\text{dr}})](\lambda), \tag{27}$$

with $b_n(\lambda) = a_n(\lambda - x/2)$ and

$$\theta_n(\lambda) = \frac{\rho_n(\lambda)}{\rho_n^t(\lambda)}. \tag{28}$$

Eqs. (21), (25), and (26) will be used to compute the observable profiles arising at late times in the bipartition protocol.

### 2.2.2 The TBA in the gapless regime

The analysis of the gapless regime follows once again the treatment of the Hamiltonian model [66]. In this case, the string centers satisfy $\lambda \in (-\infty, \infty)$, while the number and types of allowed strings depend in a non-trivial way on $\gamma$. In particular, a simplified structure is achieved at the so-called root-of-unity points, corresponding to $\gamma/\pi \in \mathbb{Q}$. In this work, we will restrict to the simplest case where $\gamma$ takes the form

$$\frac{\gamma}{\pi} = \frac{1}{p+1}, \tag{29}$$

where $p \geq 1$ is an integer. In this case, there are $N_b = p+1$ different types of strings, which we index by an integer $j \in \{1, \ldots N_b\}$. Each type is characterized by a number of quasiparticles $n_j$ and a parity $v_j = \pm 1$, taking the values

$$\begin{aligned} n_j &= j, & v_j &= 1, & j &= 1, 2 \ldots, p, \\ n_{p+1} &= 1, & v_{p+1} &= -1. \end{aligned} \tag{30}$$

Following [66], we also introduce the numbers

$$q_j = p + 1 - n_j, \quad j = 1, 2 \ldots, p, \tag{31}$$

$$q_{p+1} = -1. \tag{32}$$

The thermodynamic limit of the Bethe equations (9) then takes the form

$$\text{sign}(q_m)\big[\rho_m^t(\lambda)\big] = a_m^{(ix/2)}(\lambda) - \sum_{n=1}^{N_b}(a_{mn}*\rho_m)(\lambda),\tag{33}$$

where we used the notation (14) and where the convolution is now defined as

$$(f*g)(\lambda) := \int_{-\infty}^{\infty} d\mu\, f(\mu-\lambda)g(\mu).\tag{34}$$

In addition, we also introduced

$$a_j(\lambda) = \frac{v_j}{\pi}\frac{\sin(\gamma n_j)}{\cosh(2\lambda)-v_j\cos(\gamma n_j)} \equiv a_{n_j}^{v_j}(\lambda),\tag{35}$$

$$a_{jk}(\lambda) = (1-\delta_{n_j n_k})a_{|n_j-n_k|}^{v_j v_k}(\lambda) + 2a_{|n_j-n_k|+2}^{v_j v_k}(\lambda) + \ldots + 2a_{n_j+n_k-2}^{v_j v_k}(\lambda) + a_{n_j+n_k}^{v_j v_k}(\lambda).\tag{36}$$

As in the gapped phase, the quasi-particle rapidity distribution functions allow us to determine the effective velocities, which are obtained as the solution to the equations

$$\text{sign}(q_n)\big[\rho_n^t(\lambda)v_n^{\text{eff}}(\lambda)\big] = \frac{1}{2\pi}\text{sign}(q_n)E_n'(\lambda) - \sum_{m=1}^{N_b}\big(a_{nm}*(\rho_m v_m^{\text{eff}})\big)(\lambda),\tag{37}$$

where $E_n'(\lambda)$ will be defined later, cf. Eq. (69).

We conclude this section by reporting the expressions, valid in the gapless regime, for the expectation value of the local observables previously introduced. First, the averaged magnetization (20) can be computed via the formula

$$\overline{m} = 1 - 2\sum_{j=1}^{N_b}\int_{-\infty}^{\infty} d\lambda\, n_j\rho_j(\lambda),\tag{38}$$

which is analogous to Eq. (21). Second, for the corresponding current we have

$$\langle J_{2n}\rangle = -2\sum_{j=1}^{N_b}\int_{-\infty}^{\infty} d\lambda n_j\rho_j(\lambda)v_j^{\text{eff}}(\lambda).\tag{39}$$

Finally, the expectation value of the spin-$z$ local magnetization reads [33]

$$\langle \sigma_{2k+m}^z\rangle = 1 - 2\sum_{j=1}^{N_s}\int d\lambda\, n_j\theta_j(\lambda)\sigma_j a_j^{(m)\,\text{dr}}(\lambda),\tag{40}$$

where $\sigma_j = \text{sgn}(q_j)$, while

$$a_j^{(m)\,\text{dr}}(\lambda) = a_p(\lambda-(-1)^m\tfrac{ix}{2}) - \sum_{q=1}^{N_s}\int d\mu\, a_{jq}(\lambda-\mu)\theta_q(\mu)\sigma_q a_q^{(m)\,\text{dr}}(\mu),\tag{41}$$

and $\theta_j$ is defined in Eq. (28).

# 3 The bipartition protocol

## 3.1 The initial states

For systems with traditional Hamiltonian dynamics, most of previous work on bipartition protocols have focused on the setting where two halves of the system are prepared in different equilibrium states. Typically, the latter are chosen as finite-temperature states with different temperatures or chemical potentials [44–47, 74–85]. In this case, when the two halves are joined together, the non-trivial dynamics only takes place in a region increasing linearly in time and centered around the junction.

Alternatively, one could consider preparing two halves of the system in non-equilibrium states with a simple structure [49]. As discussed in Sec. 1, each half of the system far enough from the junction is then expected to relax to a stationary GGE which depends on the initial state. Accordingly, at late time the non-trivial dynamics again takes place only in a region increasing linearly in time and centered around the junction.

In the quantum-circuit setting under consideration, the Hamiltonian (5) is not conserved by the discrete time evolution, and the corresponding Gibbs states are not equilibrium states. Therefore, the most natural choice is to initialize the two halves of the system into different nonequilibrium states, which we choose as product states. This setting is natural in light of benchmark implementation in quantum devices, since product states are the easiest to be prepared in digital platforms.

Our analysis relies on the possibility to characterize the GGE associated to the chosen initial states. As shown in Ref. [33], this problem can be solved for a large family of states called integrable, generalizing the constructions of the Hamiltonian case [86–88]. The family of integrable states includes all two-site shift-invariant product states, allowing us to study a wide class of bipartition protocols. For concreteness, we will focus on the following states:

- the Néel state

$$|\Psi_{\mathrm{N}}\rangle = |0\rangle \otimes |1\rangle \otimes \cdots |0\rangle \otimes |1\rangle \,, \tag{42}$$

- the anti-Néel state

$$|\Psi_{\mathrm{AN}}\rangle = |1\rangle \otimes |0\rangle \otimes \cdots |1\rangle \otimes |0\rangle \,, \tag{43}$$

- the Majumdar-Ghosh or dimer state

$$|\Psi_{\mathrm{D}}\rangle = \left(\frac{|01\rangle - |10\rangle}{\sqrt{2}}\right)^{\otimes L/2} \,. \tag{44}$$

In the following, we provide analytic formulas for the rapidity distribution functions of GGEs corresponding to these states. As usual, we give a separate discussion for the gapped and gapless regime. As the derivation of these results is quite technical, we report it in Appendix B, to which the reader is referred to for further details.

### 3.1.1 Gapped phase

In the gapped phase, the GGE is characterized by an infinite set of functions $\{\eta_j\}_{j\geq 1}$, defined in Eq. (17). For the states under consideration, all those functions follow from the knowledge of $\eta_0 = 0$ and $\eta_1$ through iterated applications of the following "Y–system" relation

$$\eta_j\left(\lambda + i\frac{\gamma}{2}\right)\eta_j\left(\lambda - i\frac{\gamma}{2}\right) = \left[1 + \eta_{j+1}(\lambda)\right]\left[1 + \eta_{j-1}(\lambda)\right] \,, \tag{45}$$

where we recall that $\gamma = i\eta$ is purely imaginary throughout the gapped phase.

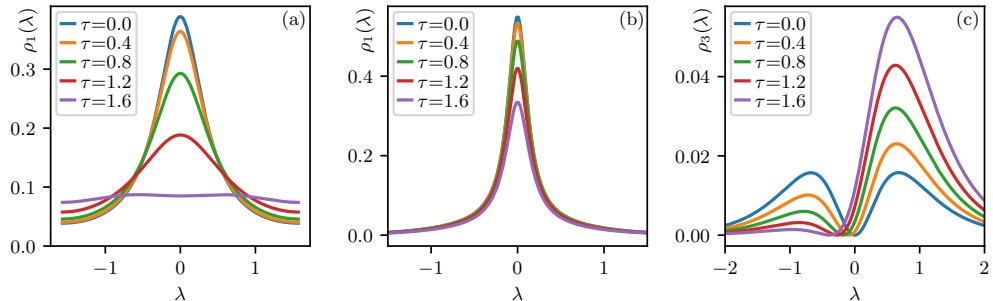

Figure 2: GGE rapidity distribution functions $\rho_n(\lambda)$ for the Néel initial state. (a): $\rho_1(\lambda)$ in the gapped phase $\Delta = 2$, and different values of $\tau$. (b), (c): same plot for $\rho_1(\lambda)$ and $\rho_3(\lambda)$, respectively, in the gapless regimes and different values of $\tau$. The value of $\Delta$ is chosen so that $\gamma = \pi/3$, cf. Eq. (7a).

The function $\eta_1$, in turn, is given by

$$1 + \eta_1(\lambda) = (1 + \mathfrak{a}(i\lambda - \gamma/2))(1 + 1/\mathfrak{a}(i\lambda + \gamma/2)) \,. \tag{46}$$

For the Néel and anti-Néel state, we find

$$\mathfrak{a}(u) = \frac{\sin(2u + \gamma)}{\sin(2u - \gamma)} \frac{\sin\left(u \pm \frac{x}{2}\right)}{\sin\left(u \mp \frac{x}{2}\right)} \frac{\sin\left(u - \gamma \pm \frac{x}{2}\right)}{\sin\left(u + \gamma \mp \frac{x}{2}\right)} \,, \tag{47}$$

while for the dimer state

$$\mathfrak{a}(u) = \frac{\tan(u + \gamma)}{\tan(u - \gamma)} \,, \tag{48}$$

see Appendix B. Note that the fact that the GGEs arising from the Néel and anti-Néel states display the same rapidity distribution functions in the gapped phase implies that they can not be distinguished by any local correlation function.

Given the functions $\eta_n(\lambda)$, the rapidity distribution functions are obtained solving Eqs. (18), which can be done by truncating the infinite system and using standard iterative methods. An example of a numerical solution is displayed in Fig. 2. In the limit $x = 0$, the rapidity distribution functions above recover the well-known results for the Hamiltonian case [68,89–94]. Note that, while the dependence in the parameter $x$ disappears in the dimer case, the GGE densities $\rho_n$ depend on it as $x$ appears in Eq. (18).

### 3.1.2 Gapless phase

As stated previously, the GGE description in the gapless phase depends in a non-trivial way on the specific value of $\gamma$. Following Section 2.2, we shall limit ourselves to the simplest case where $\gamma$ takes the form (29). The GGE is then characterized by a family of functions $\{\eta_j\}_{j=1,\dots p+1}$, which are obtained from a set of auxiliary functions $\{Y_j\}_{j=1,\dots p}$, $K$, $\mathfrak{b}$ whose construction is detailed in Appendix B. For concreteness, we give below the explicit rapidity distributions for $p = 2, 3$. For $p = 2$ and the Néel state, we have

$$\eta_1(\lambda) = \eta_2(\lambda) = \frac{\cosh(2\lambda + ix)}{\cosh(2\lambda - ix)} \,, \tag{49}$$

while for the dimer,

$$\eta_1(\lambda) = 1/\eta_2(\lambda) = (\tanh \lambda)^2 \,. \tag{50}$$

For $p = 3$, we have, for the Néel state,

$$\eta_1(\lambda) = \frac{4\sinh^2(2\lambda)(2\cos(2x)+1)}{2\cosh(4\lambda)-4\cosh(2\lambda)\cos(x)+2\cos(2x)+1},$$

$$\eta_2(\lambda) = -\frac{(1+2\cos(x-2i\lambda)))(4\cosh(2\lambda)\cos(x)-1)}{\sin^2\left(\frac{1}{2}(x+2i\lambda)\right)(4+8\cos(x+2i\lambda))},$$

$$\eta_3(\lambda) = -\frac{2(\cos(x-2i\lambda)-\cos(2(x-2i\lambda)))}{(1+2\cos(x+2i\lambda))(4\cosh(2\lambda)\cos(x)-1)},$$

(51)

and for the dimer,

$$\eta_1(\lambda) = 3\tanh^2(\lambda),$$

$$\eta_2(\lambda) = 2 - \frac{6}{1+2\cosh(2\lambda)},$$

$$\eta_3(\lambda) = \frac{1}{8}\left(4 + \frac{3}{\sinh(\lambda)^2}\right).$$

(52)

These expressions can be plugged into the Bethe equations (33) to obtain the corresponding rapidity distribution functions, see Fig. 2 for an example.

## 3.2 GHD equations

The theory of GHD describes the system in the Euler scaling limit (large times and large system sizes). In the Hamiltonian setting, its formulation is based on the TBA description of the model, from which one can read off three necessary ingredients: The quasi-momentum $P_n(\lambda)$ and the energy $E_n(\lambda)$ of each quasi-particle type, and the two-particle scattering phase shifts $\varphi_{nm}(\lambda,\mu)$ between two strings of lengths $n$, $m$ and centers $\lambda$, $\mu$, respectively.

In our work, we face the problem of generalizing the standard GHD framework to the quantum-circuit setting. To this end, it is useful to view the two-step evolution operator as generated by the Floquet Hamiltonian $H_F$, defined via

$$U(\tau) = e^{-i\tau H_F}.$$

(53)

Compared to the continuous-time setting, the Floquet Hamiltonian is not local. However, the corresponding charges are local [29], and we can repeat the reasoning leading to the derivation of the GHD equations in the continuous-time setting [49,50]. Note also that all eigenvectors of $U(\tau)$ (which are given by the Bethe Ansatz) are also eigenvectors of $H_F$ and the corresponding eigenvalues $E_n(\lambda)$ (the quasi-energies) are unique up to integer multiples of $2\pi$. However, this additive constant is immaterial, since the GHD equation only depends on the derivatives $E'_n(\lambda)$.

In summary, we can apply the standard GHD prescriptions to the quantum circuit by replacing the energy of a quasi-particle by its quasi-energy $E_n(\lambda)$. Using these definitions, the GHD equations take the standard form [49,50]

$$\partial_t \rho_n(t,x,\lambda) + \partial_x(v_n^{\text{eff}}(t,x,\lambda)\rho_n(t,x,\lambda)) = 0,$$

(54)

where the effective velocity of string with length $n$ and rapidity $\lambda$ satisfies the following self-consistency equation depending on $\rho_n(t,x,\lambda)$

$$v_n^{\text{eff}}(t,x,\lambda) = \frac{E'_n(\lambda)}{P'_n(\lambda)} + \sum_m \int d\mu \frac{\varphi_{nm}(\lambda,\mu)}{P'_n(\lambda)}\rho_m(t,x,\mu)(v_m^{\text{eff}}(t,x,\mu)-v_n^{\text{eff}}(t,x,\mu)),$$

(55)

which can be rewritten as

$$\rho_n^t(\lambda)v_n^{\text{eff}}(\lambda) = \tfrac{1}{2\pi}E_n'(\lambda) + \sum_m \int \mathrm{d}\mu\, \varphi_{nm}(\lambda,\mu)\rho_m(\mu)v_m^{\text{eff}}(\mu). \tag{56}$$

As we will discuss in the next subsection, the form of the scattering phase depends on the phase of the model and it can be read off by comparing (56) with Eqs. (19) and (37), respectively.

### 3.3 The quasi-momentum and the quasi-energy

We finally detail the identification of the quasi-momenta and quasi-energies, separating the discussion for the gapped and gapless phases.

#### 3.3.1 Gapped phase

As discussed in Sec. 2, the eigenvalues of $U(\tau)$ depend on the solutions $\{p_j\}$ to the Bethe equation (9). Denoting by $u(\{p_j\})$ the eigenvalue of $U(\tau)$ corresponding to a set $\{p_j\}_{j=1}^M$, we have

$$u(\{p_j\}) = \prod_{j=1}^M g^+(p_j)g^-(p_j), \tag{57}$$

where

$$g^+(p) = \frac{\sinh\left(p + i\frac{x}{2} - i\frac{\gamma}{2}\right)}{\sinh\left(p + i\frac{x}{2} + i\frac{\gamma}{2}\right)}, \tag{58}$$

$$g^-(p) = \frac{\sinh\left(p - i\frac{x}{2} + i\frac{\gamma}{2}\right)}{\sinh\left(p - i\frac{x}{2} - i\frac{\gamma}{2}\right)}. \tag{59}$$

This result follows from the exact solution of the model, see e.g. Ref. [33,36]. From (57), the quasienergies are expressed as a sum over the individual rapidities, namely

$$E(\{p_j\}) = \sum_j \epsilon(p_j), \tag{60}$$

where

$$\epsilon(p) = -\frac{i}{\tau}\log\frac{\sinh\left(p + i\frac{x}{2} - i\frac{\gamma}{2}\right)}{\sinh\left(p + i\frac{x}{2} + i\frac{\gamma}{2}\right)}\frac{\sinh\left(p - i\frac{x}{2} + i\frac{\gamma}{2}\right)}{\sinh\left(p - i\frac{x}{2} - i\frac{\gamma}{2}\right)}. \tag{61}$$

Note that we have divided by $\tau$, consistent with the definition of $H_{\text{F}}$ in Eq. (53). We can now write the quasienergy of a $n$-string by summing over the $n$ rapidities composing it. Recalling that in the gapped phase the physical rapidities are $\lambda_j = ip_j$, we have

$$E_n(\lambda) = \sum_{j=-\frac{n-1}{2}}^{\frac{n-1}{2}} \varepsilon(i(\lambda+j\eta)) = -\frac{i}{\tau}\log\left[\frac{\sin\left(\lambda - \frac{x}{2} + in\frac{\eta}{2}\right)\sin\left(\lambda + \frac{x}{2} - in\frac{\eta}{2}\right)}{\sin\left(\lambda - \frac{x}{2} - in\frac{\eta}{2}\right)\sin\left(\lambda + \frac{x}{2} + in\frac{\eta}{2}\right)}\right]. \tag{62}$$

Finally, taking the derivative with respect to $\lambda$, we arrive at the result

$$E_n'(\lambda) = \frac{-\sinh(n\eta)/\tau}{\cosh^2(\frac{n\eta}{2})\sin^2(\lambda - \frac{x}{2}) + \sinh^2(\frac{n\eta}{2})\cos^2(\lambda - \frac{x}{2})} - (x \to -x). \tag{63}$$

Note that, in the limit $\tau \to 0$, the above formula recovers the known expression for the continuous time evolution under the XXZ Hamiltonian

$$E_n'(\lambda) \to -\pi \sinh(\eta) a_n'(\lambda). \tag{64}$$

The quasi-momenta can be determined from the Bethe equations (18). In summary, in the gapped phase we have

$$P_n'(\lambda) = 2\pi a_n^{(x/2)}, \tag{65}$$

$$E_n'(\lambda) = 2\pi(a_n(\lambda + x/2) - a_n(\lambda - x/2))/\tau, \tag{66}$$

while the scattering phase of two strings reads

$$\varphi_{nm}(\lambda, \mu) = -a_{nm}(\lambda - \mu). \tag{67}$$

### 3.3.2 Gapless phase

The expressions in the gapless phase can be obtained from those in the gapped regime by replacing $\lambda \to -i\lambda$ and $\eta \to i\gamma$. Taking into account the sign $\mathrm{sgn}(q_n)$ appearing in the Bethe equations, we recover the standard formulation of GHD by identifying

$$P_n'(\lambda) = 2\pi \,\mathrm{sgn}(q_n) a_n^{(ix/2)}, \tag{68}$$

$$E_n'(\lambda) = 2\pi \frac{\mathrm{sgn}(q_n)}{\tau} \left[ a_n\left(\lambda - i\frac{x}{2}\right) - a_n\left(\lambda + i\frac{x}{2}\right) \right], \tag{69}$$

while the scattering phase is

$$\varphi_{nm}(\lambda, \mu) = -\mathrm{sgn}(q_n) a_{nm}(\lambda - \mu). \tag{70}$$

## 4 Results

Having derived the GHD equations for our model, we now proceed to study them. First, following the standard theory [49, 50], we recall that the GHD equations can be written as differential equations in terms of the rescaled variable

$$\zeta = x/t, \tag{71}$$

namely

$$\zeta \partial_\zeta \rho_n(\zeta, \lambda) = \partial_\zeta (v_n^{\mathrm{eff}}(\zeta, \lambda) \rho_n(\zeta, \lambda)). \tag{72}$$

Eq. (72) describes how the local, space-time dependent GGE changes as we vary the velocity $\zeta$ of the reference frame. In the bipartition protocols, this equation is supplemented by the boundary conditions

$$\lim_{\zeta \to \infty} \rho_n(\zeta, \lambda) = \rho_n^R(\lambda), \tag{73a}$$

$$\lim_{\zeta \to -\infty} \rho_n(\zeta, \lambda) = \rho_n^L(\lambda), \tag{73b}$$

where $\rho_n^L(\lambda)$, $\rho_n^R(\lambda)$ are the GGE distribution functions associated with the left and right initial states. For the product states considered in this work, these functions were computed in Sec. 3. The above boundary conditions encode the fact that, away from the junction point, the system will locally equilibrate to a homogeneous GGE, determined by the initial states.

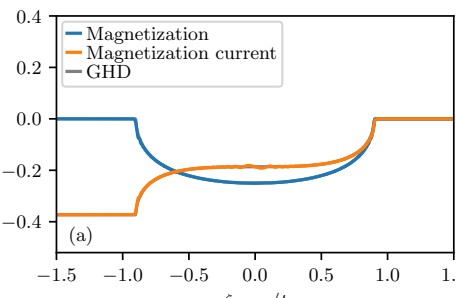
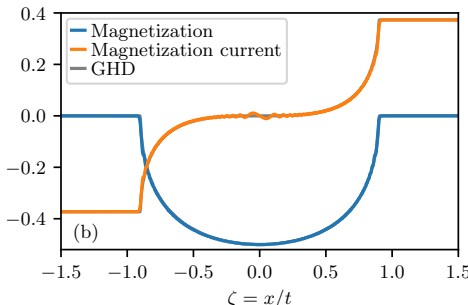

Figure 3: Late time profiles after bipartition protocols at the non-interacting points, $\tau = \pi/2, \Delta = 4.0$. (a): the left and right states are the Néel and dimer states, respectively. The plot shows the profiles for the local magnetization, $\overline{m}_j = (\langle \sigma_j^z \rangle + \langle \sigma_{j+1}^z \rangle)/2$, and the corresponding current $\langle J_j \rangle$. Gray lines are the solution to the GHD equations, while blue and orange lines correspond to the exact simulation of the dynamics using the Gaussian formalism. The latter displays small ripples, which are finite-time effects. (b): same plot, for the protocol in which the left and right states are the Néel and anti-Néel states. All profiles of the Gaussian numerical computations are obtained at time $t = 256\tau$ and were carried out on a sufficiently large system of size $L = 4096$ (note that, due to the strict lightcone on a quantum circuit, this is equivalent to simulating the infinite system).

Solving (72) with the boundary conditions (73) can be done with standard iterative methods, see e.g. Ref. [95]. In this work, we use a different numerical approach which was recently introduced in Ref. [96] and briefly explained in Appendix A. In the rest of this section we present the final results of our numerical computations. Note that the solution to (72) gives us a family of rapidity distribution functions $\{\rho_n(\zeta, \lambda)\}_\zeta$. We use the latter to compute the space-time profiles of local observables, using the formulas presented in Sec. 3.

## 4.1 The non-interacting points

We begin by discussing the non-interacting points of the model, where we can test our predictions against independent, numerically exact results. These points correspond to either one of the following choices of the circuit parameters [33]:

(i) $\Delta \neq 0$ and $\tau = 2\pi n/\Delta$, with $n \in \mathbb{Z}$;

(ii) $\Delta = 0$ and $\tau \in \mathbb{R}_+$.

At these special values of the parameter space, we can make use of the Jordan-Wigner transformation

$$c_n^\dagger = \frac{\sigma_n^x + i\sigma_n^y}{2} \prod_{j=1}^{n-1} \sigma_j^z, \qquad c_n = \frac{\sigma_n^x - i\sigma_n^y}{2} \prod_{j=1}^{n-1} \sigma_j^z, \tag{74}$$

to rewrite

$$U_o(\tau) = \exp\left[-i\frac{\tau}{2} \sum_{n=1}^{L/2} \left(c_{2n}^\dagger c_{2n+1} + c_{2n+1}^\dagger c_{2n}\right)\right], \tag{75}$$

$$U_e(\tau) = \exp\left[-i\frac{\tau}{2} \sum_{n=1}^{L/2} \left(c_{2n+1}^\dagger c_{2n+2} + c_{2n+2}^\dagger c_{2n+1}\right)\right], \tag{76}$$

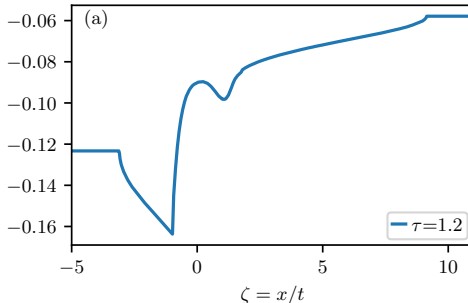
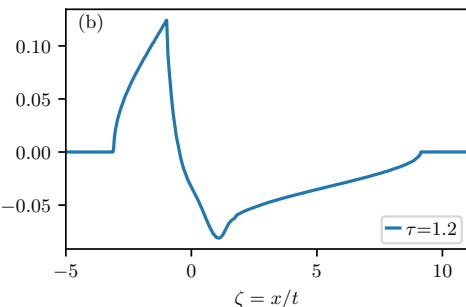

Figure 4: Late time profiles after the bipartition protocol in which the left and right states are the Néel and dimer states, respectively. The plots correspond to (a) the local staggered magnetization $(\langle\sigma_j^z\rangle-\langle\sigma_{j+1}^z\rangle)/2$ and (b) the averaged local magnetization $\overline{m}_j = (\langle\sigma_j^z\rangle + \langle\sigma_{j+1}^z\rangle)/2$. We set $\tau = 1.2$, while the value of $\Delta$ is chosen in such a way that $\gamma = \pi/3$.

where $\{c_n\}$, $\{c_n^\dagger\}$ satisfy the canonical anti-commutation relations.

Eqs. (75) and (76) make it explicit that the operators $U_e(\tau)$, $U_o(\tau)$ are written as the exponential of Hamiltonians which are quadratic in the fermion operators and thus non-interacting. As we explain in Appendix C, the corresponding dynamics can then be computed in a numerically-exact and efficient way by means of the so-called Gaussian formalism [97], allowing us to produce data for large system sizes and simulation times. These computations serve as an important benchmark for the GHD equations and our numerical solution to them. As we discuss in the following, the GHD predictions are always found to be in perfect agreement with the exact numerics, up to controlled and expected finite-size and finite-time effects.

In Fig. 3(a), we first show data for the protocol where the left and right halves of the system are initialized in the Néel and dimer state respectively. The plot shows the profiles of the local magnetization and the corresponding current. We see that the profiles are flat outside the "lightcone" $[-\zeta_{\max}, \zeta_{\max}]$, which is an expected feature: indeed $\zeta_{\max}$ corresponds to the maximum velocity at which quasi-particles propagate. Taking reference frames moving at a velocity with larger absolute value, the system is described by the GGEs associated by the initial left/right states.

The magnetization is vanishing at $\zeta = \pm\infty$, but the profiles is non-trivial inside of the lightcone, signaling the emergence of non-trivial, non-equilibrium steady states. We also note that the magnetization and current profiles display no points of non-analiticities inside of the lightcone. This is expected, since at the non-interacting points we have a single quasi-particle string type [47].

The features described above are qualitatively similar to what one could have expected based on the GHD in the XXZ Heisenberg model [49,53]. Yet, away from the limit of small Trotter step, one may expect the emergence of qualitative differences compared to the continuous-time evolution [33]. We can see that this is indeed the case even at the non-interacting point of the model with non-zero $\tau$. To this end, we consider the bipartition protocol where the left and right states are prepared in a Néel and anti-Néel state, respectively. Note that the resulting initial state can be thought of as a homogeneous Néel state in which a defect, in the form of a $|0\rangle$ qubit, is added in the center of the chain.

This is a peculiar situation because the Néel and anti-Néel states are related by a translation of a single site. In models in which all charges satisfy single-site translation symmetry, such as the XXZ Heisenberg chain, both states equilibrate to the same GGE [98]. Accordingly, after an initial transient time, there would be no non-trivial dynamics at large space-time scales. This is intuitive when viewing the initial state as created by the addition of a single

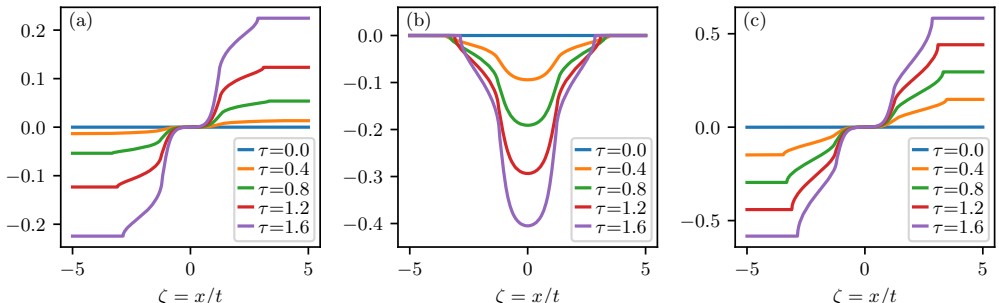

Figure 5: Late time profiles after the bipartition protocol in which the left and right states are the Néel and anti-Néel states. The plots show (a) the local staggered magnetization $(\langle\sigma_j^z\rangle - \langle\sigma_{j+1}^z\rangle)/2$, (b) the averaged local magnetization $\overline{m}_j = (\langle\sigma_j^z\rangle + \langle\sigma_{j+1}^z\rangle)/2$, and (c) the current $\langle J_j\rangle$. Each curve corresponds to a different $\tau$, while the value of $\Delta$ is always chosen in such a way that $\gamma = \pi/3$.

defect in an homogeneous configuration. Indeed, a local initial perturbation does not typically lead to a change in the macroscopic state of the system, although exceptions exist, see e.g. [99–104]. On the other hand, the Trotterized XXZ circuit is invariant under translation by two sites, which allows for the possibility that the Néel and anti-Néel states give rise to different GGEs. It is worth mentioning that one could construct local integrable Hamiltonians breaking the single-site translation symmetry (but preserving two-site shift invariance) for which a similar behavior is observed. However, while this construction would appear rather *ad hoc* in the continuous-time limit, here it appears naturally as a consequence of increasing the Trotter step. The ability to change the global late-time behavior by a single defect in a circuit setting is also of obvious interest for benchmark experiments in current digital devices. Indeed, approximating the continuous-time evolution of a two-site shift-invariant Hamiltonian would require implementing very deep circuits, even to reach short times, which is unfeasible on current devices. Instead, existing experiments [40–42] have already shown that integrable circuits can be implemented keeping sufficient coherence over a number of discrete steps that is enough to appreciate the emergence of late-time features.

Fig. 3(b) shows our quantitative results for this bipartition protocol in the non-interacting case. The profiles of the local magnetization and current demonstrate the emergence of nontrivial dynamics even at macroscopic scales. Interestingly, we see that there is an accumulation of the magnetization charge inside of the lightcone, with the current changing sign at the left and right ends. This does not violate the global conservation of the $z$-magnetization in the system, as the excess charge propagates to infinity. It is natural to ask whether this peculiar behaviour is a special property of the non-interacting points. We see in the next section that this is not the case, and that non-trivial dynamics in this bipartition protocol survives the presence of interactions.

## 4.2 Observable profiles at the interacting points

We finally present our results for different bipartition protocols at generic points of the parameter space.

First, it is important to state that, for the initial states considered, the magnetization and current profiles are trivial in the gapped regime. This is because both the Néel and dimer states have vanishing magnetization, and in the gapped regime there are no additional charges breaking the spin-flip symmetry [33]. We have numerically verified this prediction by solving the GHD equations and computing the magnetization and current profiles. In all cases, we have

found that the profiles of these observables are flat, up to the chosen numerical accuracy. Note, that the solutions $\{\rho_n(\zeta,\lambda)\}$ to the GHD equations for the Néel-dimer bipartition protocol do have a non-trivial dependence on $\zeta$, even in the gapped regime. Conversely, the GGEs associated with the Néel and anti-Néel states coincide, and there is no non-trivial dynamics taking place in the corresponding bipartition protocol.

Next, we move on to discuss the gapless phase. In Fig. 4, we study the protocol where the left and right states are initialized in the dimer and Néel states, respectively. We plot the profiles of the staggered and averaged magnetization, respectively, both showing non-trivial dynamics. Compared to the non-interacting circuit, we see additional points inside of the lightcone where the profiles are not smooth. Similarly to the GHD profiles in the XXZ Heisenberg chain [49,53], these points are associated to the maximum velocities of distinct quasi-particle types (namely, of strings with increasing length $n$). We also see that both observables display non-monotonic and highly irregular profiles. While the figure shows data for one value of $\tau$ and $\Delta$, we have verified that qualitatively similar features arise for different choices of the circuit parameters.

Finally, we discuss the bipartition protocol where the two halves of the system are prepared in the Néel and anti-Néel state, respectively. We report an example of our results in Fig. 5, where we plot the profiles of the staggered and averaged magnetization, and the current (again in the gapless phase). As anticipated, all plots show non-trivial profiles and, once again, we see non-analyticity points inside of the lightcone, although they are less evident than in the Néel-dimer bipartition protocol. As in the non-interacting case, we see that the dynamics causes an accumulation of magnetization at the origin [Fig. 5(b)], with the current taking opposite values at the left and right ends of the lightcone.

# 5 Outlook

In this work, we have developed and studied the GHD of integrable quantum circuits. Our main finding is that, for generic choices of the circuit parameters, the large-scale dynamics displays qualitative differences compared to the continuous-time evolution of the corresponding XXZ Hamiltonian. In particular, considering quenches from product states, we predict peculiar features which can be detected by looking at the late-time dynamics of simple observables, such as the local magnetization.

Our work opens several directions for future research. First, it would be important to provide predictions for more general observables and correlations than those considered in this work. Technically, this problem requires the derivation of new formulas for the expectation value of local observables in stationary states described by a given rapidity distribution function. While this is a hard problem, we expect that one could generalize the results available for the Hamiltonian model [67–70], extending the computation presented in Ref. [33] for the simplest case of one-point functions.

Along similar lines, it would be interesting to also consider the entanglement dynamics in integrable quantum circuits. In the Hamiltonian setting, building upon the results for the evolution of bipartite entanglement in homogeneous systems [105], recent work has shown how the GHD formalism is naturally suited to describe the entanglement dynamics in inhomogeneous setting [47,55]. We expect that the results of Refs. [105] apply directly to integrable quantum circuits, potentially allowing us to investigate the role of discrete space-time in the large-scale dynamics of bipartite entanglement.

Finally, it would be important to analyze the feasibility of observing some of the features predicted by our work in benchmark experiments in existing quantum platforms [40–42]. To this end, it would be necessary to study how finite-size corrections affect our predictions. In addition, one would also need to analyze the effect of noise, possibly taking into account recent

strategies for quantum error-mitigation [106–115]. We leave these interesting questions for future work.

## Acknowledgments

We acknowledge helpful discussions with Bruno Bertini and especially Benjamin Doyon for early collaboration on this project.

**Funding information** FH acknowledges funding from the faculty of Natural, Mathematical & Engineering Sciences at King's College London. The CREATE cluster [116] was used for numerical simulations. This work was co-funded by the European Union (ERC, QUANTHEM, 101114881). Views and opinions expressed are however those of the author(s) only and do not necessarily reflect those of the European Union or the European Research Council Executive Agency. Neither the European Union nor the granting authority can be held responsible for them.

## A Numerical solution of the GHD equation

We solve the GHD equation numerically using a recently introduced fixed point equation [96]. Given the states on the left ($-$) and the right side ($+$) either in terms of their quasi-particle densities $\rho_{\pm;i}(\lambda)$ or equivalently as occupation functions $\theta_{\pm;i}(\lambda)$, we can compute

$$P'^{\mathrm{dr}}_{\pm;i}(\lambda) = P'_i(\lambda) + \sum_j \int \mathrm{d}\mu\, \varphi_{ij}(\lambda - \mu)\rho_{\pm;j}(\mu). \tag{A.1}$$

Then define:

$$K_{0;i}(x,\lambda) = P'^{\mathrm{dr}}_{\mathrm{sgn}(x),i}(\lambda)x, \tag{A.2}$$

$$\hat{\theta}_{0;i}(k,p) = \theta_{\mathrm{sgn}(k);i}(\lambda), \tag{A.3}$$

$$\hat{N}_{0;i}(k,p) = \theta_{\mathrm{sgn}(k);i}(\lambda)k. \tag{A.4}$$

As derived in Ref. [96], $x \to K_{0;i}(x,\lambda)$ can be interpreted as a change of spatial coordinates system that trivializes the GHD equation, i.e. $\partial_t \hat{\theta}_i(k,\lambda) = 0$. To obtain the solution at space-time point $(t,x)$, we thus only need to find $K_i(t,x,\lambda)$, which satisfies the following integral-fixed-point equation [96]:

$$
\begin{aligned}
K_i(t,x,\lambda) &= P'_i(\lambda)x - E'_i(\lambda)t + \hat{\mathbf{T}}\hat{N}_{0;j}(K_j(t,x,\mu),\mu) \\
&= P'_i(\lambda)x - E'_i(\lambda)t + \hat{\mathbf{T}}[\theta_{\mathrm{sgn}(K_j(t,x,\mu));j}(\mu)K_j(t,x,\mu)].
\end{aligned} \tag{A.5}
$$

Here $\hat{\mathbf{T}}f_i(\lambda) = \sum_j \int \mathrm{d}\mu\, \varphi_{ij}(\lambda,\mu)f_j(\mu)$. Once the solution $K_i(t,x,\lambda)$ is obtained we can compute the solution to the GHD equation as follows:

$$\theta_i(t,x,\lambda) = \hat{\theta}_i(K_i(t,x,\lambda),\lambda) = \theta_{\mathrm{sgn}(K_i(t,x,\lambda));i}(\lambda). \tag{A.6}$$

We solve Eq. (A.5) by iteration with initial guess $P'_i(\lambda)x - E'_i(\lambda)t$, which converges in the gapless phase, where only finitely many strings are present.

## A.1 Upgraded algorithm

As it turns out for the Néel and Dimer state in the gapped phase this iterative scheme does not converge. The same holds for the dressing equation (which can be viewed as a linearized version of (A.5)). The iteration scheme does not converge because $n_i(\lambda)$ does not decay sufficiently fast for large strings $i \to \infty$: Consider the first term of the iteration procedure to compute $f_i^{\mathrm{dr}}(\lambda)$

$$f_i^{\mathrm{dr}}(\lambda) = f_i(\lambda) + \hat{\mathbf{T}}\theta_j(\mu)f_j(\mu) + \dots \qquad (A.7)$$

Since $n_i(\lambda)$ does not decay fast enough, already the first term $\hat{\mathbf{T}}n_j(\mu)f_j(\mu)$ diverges, unless $f_i(\lambda)$ decays sufficiently fast (which it does not for instance for $f_i(\lambda) = P'_i(\lambda)$). We can however slightly modify the iteration procedure to obtain a convergent fixed-point iteration.

Let us first explain the idea starting with the dressing equation:

$$f_i^{\mathrm{dr}}(\lambda) = f_i(\lambda) + \hat{\mathbf{T}}\theta_j(\mu)f_j^{\mathrm{dr}}(\mu). \qquad (A.8)$$

We can rewrite this as follows: Choose any functions $g_i(\lambda)$ and $m_i(\lambda)$ and define

$$b_i(\lambda) = g_i(\lambda) - \hat{\mathbf{T}}m_j(\mu)g_j(\mu). \qquad (A.9)$$

Now write $f_i^{\mathrm{dr}}(\lambda) = w_i(\lambda)g_i(\lambda)$ and observe:

$$w_i(\lambda)g_i(\lambda) = f_i(\lambda) + \hat{\mathbf{T}}\theta_j(\mu)w_j(\lambda)g_j(\lambda), \qquad (A.10)$$

$$w_i(\lambda)g_i(\lambda) = w_i(\lambda)b_i(\lambda) + w_i(\lambda)\hat{\mathbf{T}}m_j(\mu)g_j(\mu). \qquad (A.11)$$

Combining both equations we find:

$$w_i(\lambda) = \frac{1}{b_i(\lambda)}\left(f_i(\lambda) + \hat{\mathbf{T}}\theta_j(\mu)g_j(\lambda)w_j(\lambda) - w_i(\lambda)\hat{\mathbf{T}}m_j(\mu)g_j(\mu)\right). \qquad (A.12)$$

This is a new fixed point equation for $w_i(\lambda)$. It is motivated by the equation for the effective velocity, which is obtained in the special case $f_i(\lambda) = E'_i(\lambda)$, $b_i(\lambda) = P'_i(\lambda)$ and $m_i(\lambda) = n_i(\lambda)$. Intuitively speaking we need that $\zeta_i(\lambda)$ decays fast for large $i$, but at the same time $b_i(\lambda)$ should decay slowly, such that (A.12) converges.

For the gapped phase we used $g_i(\lambda) = 1/2^i$ and $m_i(\lambda) = 1$, which happened to make (A.12) convergent with initial guess $f_i(\lambda)/b_i(\lambda)$.

Similarly, one can derive an equation for $\tilde{K}_i(t,x,\lambda) = K_i(t,x,\lambda)/g_i(\lambda)$:

$$\tilde{K}_i(t,x,\lambda) = \frac{1}{b_i(\lambda)}\left(P'_i(\lambda)x - E'_i(\lambda)t - \tilde{K}_i(t,x,\lambda)\hat{\mathbf{T}}m_j(\mu)g_j(\mu) + \hat{\mathbf{T}}[\hat{N}_{0;j}(g_j(\mu)\tilde{K}_j(t,x,\mu),\mu)]\right), \qquad (A.13)$$

which for the partitioning protocol becomes, for $g_i(\lambda) > 0$:

$$\tilde{K}_i(t,x,\lambda) = \frac{1}{b_i(\lambda)}\Big(P'_i(\lambda)x - E'_i(\lambda)t - \tilde{K}_i(t,x,\lambda)\hat{\mathbf{T}}m_j(\mu)g_j(\mu)$$
$$+ \hat{\mathbf{T}}[\theta_{\mathrm{sgn}(\tilde{K}_j(t,x,\mu));j}(\mu)g_j(\mu)\tilde{K}_j(t,x,\mu)]\Big). \qquad (A.14)$$

This equation, with the above stated $g_i(\lambda) = 1/2^i$ and $m_i(\lambda) = 1$, is solved by iteration, starting from $\frac{P'^{\mathrm{dr}}_{\mathrm{sgn}(x),i}(\lambda)}{b_i(\lambda)}x$, to obtain the solution to the GHD equation in the gapped case.

# B Derivation of the GGE rapidity distribution functions

In this section we describe the derivation of the GGE rapidity distribution functions for the quasiparticles and holes $\rho_n(\lambda)$, $\rho_n^h(\lambda)$, which emerges after local relaxation from the initial states involved in our bipartition protocol.

The derivation follows from the boundary quantum transfer matrix (BQTM) approach explained in Ref. [86] for the case of a continuous Hamiltonian evolution, see also [87, 88, 117–119], and extended in [33] to the circuit dynamics.

## B.1 The boundary quantum transfer matrix construction

Our starting point is a continuous family of initial states

$$|\Psi_0(z)\rangle = \left( \frac{z^{1/2}|01\rangle - z^{-1/2}|10\rangle}{\sqrt{|z| + |z|^{-1}}} \right)^{\otimes L/2}, \tag{B.1}$$

which interpolates between the various initial states considered in our partitioning protocol. More concretely, Néel, anti-Néel and dimer states are recovered for $z = 0$, $z = \infty$ and $z = 1$ respectively.

The BQTM approach allows to compute the GGE rapidity distributions from initial states of the form (B.1) by recasting the discrete time evolution in terms of auxiliary transfer matrices acting in the space direction, and where the initial state enters as a boundary condition.

We refer to the previous works [33, 86–88, 117] for a complete definition of the boundary QTM. Its boundaries are specified by a set of parameters, which in the present case are given by

$$\xi_{\pm} = \arctan\left( \frac{z+1}{z-1} \tan\left( \frac{x}{2} \pm \frac{\gamma}{2} \right) \right) \tag{B.2}$$

(as oppposed to the case of most generic two-site product states considered in [87], the states (B.1) correspond to diagonal boundary conditions, so all the other parameters vanish).

In the case of Hamiltonian evolution, the BQTM was originally introduced to compute a quantity called the Loschmidt echo [87, 117], resulting in a transfer matrix acting on $N$ auxiliary spins (where $N$ is a introduced as a Trotterization number for the time evolution). Physical quantities of interest were then related to the dominant eigenvalue of this auxiliary transfer matrix, expressed in terms of a set of auxiliary Bethe roots. In particular, the GGE rapidity distributions are obtained by taking a "zero time limit" in the BQTM generating the Loschmidt echo. In this limit, the auxiliary Bethe roots collapse to some known analytical value and their contribution to the dominant eigenvalue cancels. As a result, the GGE rapidity distributions can then be simply recovered from a BQTM with $N = 0$ sites, namely a scalar. The same observations hold in the circuit case (where a BQTM with $N = 2$ was used in our previous work [33]), and in this work we shall therefore work from the start with the $N = 0$ version of the BQTM, which reads

$$\mathcal{T}(u) = \omega_1(u) + \omega_2(u), \tag{B.3}$$

where

$$\omega_1(u) = \frac{\sin(2u + \gamma)\sin\left(u + \xi^+ - \frac{\gamma}{2}\right)\sin\left(u + \xi^- - \frac{\gamma}{2}\right)}{\sin(2u)},$$
$$\omega_2(u) = \frac{\sin(2u - \gamma)\sin\left(u - \xi^+ + \frac{\gamma}{2}\right)\sin\left(u - \xi^- + \frac{\gamma}{2}\right)}{\sin(2u)}. \tag{B.4}$$

## B.2 The T–system and Y–system

The (scalar) transfer matrices $\mathcal{T}(u)$ are part of a hierarchy of commuting transfer matrices $\mathcal{T}_i(u)$, satisfying the recursive relations [87]

$$
\begin{aligned}
\mathcal{T}_0 &= 1\,, \\
\mathcal{T}_1 &= \mathcal{T}\,, \\
\mathcal{T}_j &= \mathcal{T}_{j-1}^- \mathcal{T}_1^{[j-1]} - f^{[j-3]}\mathcal{T}_{j-2}^{[-2]}\,, \qquad j \geq 2\,,
\end{aligned}
\tag{B.5}
$$

where for any function $F(u)$ we have introduced the short-hand notation $F^{[k]}(u) \equiv F(u+ik\gamma/2)$, and where $f(u-\gamma/2) = \omega_1(u+\gamma/2)\omega_2(u-\gamma/2)$. In the more general case where the transfer matrix acts on a chain of $N$ spins, or for more general (e.g. non-diagonal) boundary conditions, the definition of $f$ is changed, but the general structure (B.5) remains the same [87].

The transfer matrices $\mathcal{T}_j$ satisfy a set of relations known as the T–system [87, 120, 121], and which read

$$
\mathcal{T}_j^{[m]}\mathcal{T}_j^{[-m]} = \mathcal{T}_{j+m}\mathcal{T}_{j-m} + \Phi_{j-m+1}\mathcal{T}_{m-1}^{[j+1]}\mathcal{T}_{m-1}^{[-j-1]}\,,
\tag{B.6}
$$

where we have introduced the function $\Phi_j = \prod_{l=1}^{j} f^{[2k-2-j]}$.

One then introduces the family of Y–functions $Y_0 = 0$, and

$$
Y_j(\lambda) = \frac{\mathcal{T}_{j-1}(u)\mathcal{T}_{j+1}(u)}{\Phi_j(u)}\,, \qquad j \geq 1\,.
\tag{B.7}
$$

As a consequence of the T–system, the Y–functions obey the following set of relations, known as Y–system [120]

$$
Y_j\left(u + \frac{\gamma}{2}\right)Y_j\left(u - \frac{\gamma}{2}\right) = \left[1 + Y_{j+1}(u)\right]\left[1 + Y_{j-1}(u)\right]\,,
\tag{B.8}
$$

and can therefore be obtained recursively from the knowledge of $Y_1(\lambda)$, which is

$$
1 + Y_1(u) = (1 + \mathfrak{a}(u - \gamma/2))(1 + 1/\mathfrak{a}(u + \gamma/2))\,,
\tag{B.9}
$$

where

$$
\mathfrak{a}(u) = \frac{\omega_1(u)}{\omega_2(u)} = \frac{\sin(2u+\gamma)}{\sin(2u-\gamma)}\frac{\sin(u+\xi_+ - \gamma/2)\sin(u+\xi_- - \gamma/2)}{\sin(u-\xi_+ + \gamma/2)\sin(u-\xi_- + \gamma/2)}\,.
\tag{B.10}
$$

Note that for the dimer state, $z = 1$, the dependence in $x$ vanishes (which can already be noted from the fact that $\xi_\pm$ do not depend on $x$).

As shown in previous works, the GGE functions $\eta_j(\lambda)$ are then obtained from the Y–functions [87, 88]. For generic values of the anisotropy $\Delta$, in particular in the gapped phase $|\Delta| > 1$ (that is, $\gamma = i\eta$, $\eta \in \mathbb{R}$), the GGE is characterized by an infinite family $\{\eta_j(\lambda)\}_{j\geq 1}$, which are simply identified one by one with the $Y$ functions, i.e. $\eta_j(\lambda) = Y_j(\lambda)$.

Things are however more subtle in the gapless phase $|\Delta| < 1$, which is spanned by the so-called root-of-unity points $\gamma/\pi \in \mathbb{Q}$: for such values of $\gamma$ the TBA truncates to a finite number of rapidity distribution functions $\{\eta_j(\lambda)\}_{j=1,\dots N_b}$. Accordingly, the T– and Y– system can also be truncated to a finite number of equations, and we now turn to a description of the precise correspondence between the two, limiting ourselves, for simplicity, to values of the anisotropy of the form $\gamma = \frac{\pi}{p+1}$, with $p$ an integer $\geq 1$.

## B.3 Truncation of the T– and Y– system at $\gamma = \frac{\pi}{p+1}$

At $\gamma = \frac{\pi}{p+1}$ there exists an additional linear relation between the transfer matrices. It generally takes the form [88]

$$\mathcal{T}_{p+1}(u) = \alpha(u)\mathcal{T}_{p-1}(u) + \beta(u), \tag{B.11}$$

where, in the present case, the functions $\alpha(u)$ and $\beta(u)$ are given by

$$\alpha(u) = \frac{\left(\Omega^{[-p-2]}\Omega_1^{[-p-2]}\Omega_2^{[-p-2]}\right)^2}{\Phi_p}, \tag{B.12}$$

$$\beta(u) = \Omega^{-[p+2]}\left((\Omega_1^{[-p-2]})^2 + (\Omega_2^{[-p-2]})^2\right), \tag{B.13}$$

and where we have defined

$$\Omega(u) = \cot((p+1)u)\cot((p+1)(u+\pi/2)),$$
$$(\Omega_{1,2}(u))^2 = \frac{\cos\left((p+1)(u\pm\xi^+)\right)\cos\left((p+1)(u\pm\xi^-)\right)}{2^{2p}}. \tag{B.14}$$

As a result of (B.11), the Y–system can also be truncated: introducing the function

$$K(u) = \frac{\Omega^{[-p-2]}\Omega_1^{[-p-2]}\Omega_2^{[-p-2]}}{\Phi_p}\mathcal{T}_{p-1}(u), \tag{B.15}$$

we have

$$Y_j^+ Y_j^- = (1 + Y_{j+1})(1 + Y_{j-1}), \quad j = 1,\dots p-1,$$
$$1 + Y_{p-1} = K^+ K^-,$$
$$1 + Y_p = 1 + (\mathfrak{b} + \mathfrak{b}^{-1})K + K^2, \qquad \mathfrak{b} = \frac{\Omega_1^{[-p-2]}}{\Omega_2^{[-p-2]}}. \tag{B.16}$$

## B.4 The GGE densities at $\gamma = \pi/(p+1)$

Using classic techniques, the truncated Y–system can be turned into a set of non-linear integral equations (NLIE) [88]. Those can in turn be compared with the generic structure of the NLIE obeyed by the functions $\eta_j$ involved in the TBA solution, which can be found in Section 9.2 of [66] for the gapless XXZ Hamiltonian at finite temperature, and retain the same general structure for Generalized Gibbs Ensembles or for the discrete dynamics (only the explicit form of source terms gets modified). In order to match the form of the NLIE inherited from the Y–system with those of the TBA, we are led to the following identification

$$\eta_j(\lambda) = Y_j(i\lambda), \qquad j = 1,\dots p-1,$$
$$\eta_p(\lambda) = \mathfrak{b}(i\lambda)K(i\lambda), \tag{B.17}$$
$$\eta_{p+1}(\lambda) = \mathfrak{b}(i\lambda)/K(i\lambda).$$

For concreteness, we give in the main text the explicit expressions of the functions $\eta_j(\lambda)$ for $p = 2$ and $p = 3$, see Eqs. (52) and (51).

## C  Details on the free-fermion points of the model

In this Appendix we provide details on the techniques we have used to study the non-interacting points of the model, namely those in which the circuit can be mapped onto the Floquet dynamics of free fermions.

From Eqs. (75) and (76), we see that the unitary operators $U_e(\tau)$, $U_o(\tau)$ are written as the exponential of quadratic Hamiltonians. This makes it possible to diagonalise the Floquet operator $U(\tau)$ by means of an appropriate Bogoliubov rotation [33]. In addition, the mapping to free fermions makes it possible to simulate efficiently the dynamics of the system, using the formalism of Gaussian states [97]. We used this method to produce the numerical data presented in Sec. 4.1, as we explain in the following.

We recall that Gaussian states are defined by the fact that they satisfy Wick's theorem [97]. Accordingly, given a pure Gaussian state $|\psi\rangle$ with fixed particle number, it is fully determined by its covariance matrix

$$\Gamma_{ij} = \langle\psi|c_i^\dagger c_j|\psi\rangle. \tag{C.1}$$

This observation drastically simplifies the problem, as $\Gamma$ is a matrix whose linear dimensions grow linearly, not exponentially, with the system size.

The unitary updates of the system state $|\psi_{2t+1}\rangle = U_o(\tau)|\psi_{2t}\rangle$ and $|\psi_{2t}\rangle = U_e(\tau)|\psi_{2t-1}\rangle$ correspond to a change in the covariance matrix which can be written as [97]

$$\Gamma(2t+1) = V_o^*\Gamma(2t)V_o^T, \tag{C.2}$$

$$\Gamma(2t) = V_e^*\Gamma(2t-1)V_e^T, \tag{C.3}$$

where $(\cdot)^*$ and $(\cdot)^T$ denote complex conjugation and transposition, respectively. Here,

$$V_e = e^{A_e}, \quad V_o = e^{A_o}, \tag{C.4}$$

where $A_e$ and $A_o$ are $L \times L$ matrices whose non-zero elements are defined by

$$(A_o)_{2n,2n+1} = (A_o)_{2n+1,2n} = -i\frac{\tau}{2}, \tag{C.5}$$

$$(A_e)_{2n-1,2n} = (A_e)_{2n,2n-1} = -i\frac{\tau}{2}. \tag{C.6}$$

We can also map the initial states (42)–(44) into fermionic Fock states, which are Gaussian. This is true also for bipartite initial states, where the two halves of the system are prepared in different product states chosen among (42)–(44). In fact, it is straightforward to compute their covariance matrix. Given the initial matrix $\Gamma$, the evolution is computed numerically via iteration of Eqs. (C.2).

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
