# Peer review of "Generalized hydrodynamics of integrable quantum circuits"

_SciPost Physics, doi:SciPost Phys. 18, 135 (2025)_

## Round 3 · Referee Report · Anonymous (Referee 1) · 2024-10-30

Report

The paper by Hübner et al. presents an interesting study of the nonequilibrium dynamics of a quantum spin chain, at the interface of three research areas that have received significant attention in recent years: (i) Trotter transitions, that is, qualitative changes in the dynamics that occur when the Trotter step size crosses a critical value; (ii) integrability and generalized hydrodynamics (GHD); (iii) quench dynamics, in particular, the bipartition protocol, which can lead to the emergence of nonequilibrium steady states. This study builds on and complements a previous work [33] that includes two of the authors of the present manuscript. The model system under investigation is an integrable quantum circuit that represents a Trotterization of the dynamics generated by the XXZ Hamiltonian. As initial states, the authors consider a single domain wall between the two halves of the spin chain in Néel, anti-Néel, or Majumdar-Ghosh states. To describe the dynamics in the space-time scaling limit, the authors build on recent developments in the theory of integrable quantum circuits, and derive the GHD equations for such systems. Apart from these technical developments, the key physical result of the paper is that for finite Trotter step sizes, the nonequilibrium macrostates appearing at late times can differ qualitatively from the continuous-time limit.

As stated above, the paper addresses a timely topic at the interface between different research areas. The results are interesting and clearly presented. I am not an expert on integrability and GHD, and, therefore, I find it hard to judge whether the paper is groundbreaking in terms of developing new techniques. However, my impression is that the foundations for the present work were laid in Ref. [33], which diminishes the claims to novelty of this work. Therefore, I believe that the paper is better suited for a less selective journal such as SciPost Physics Core.

Requested changes

1- As the authors explain, a domain wall between the Néel and anti-Néel states can be interpreted as a single localized defect, and such a defect does typically not lead to a change in the macroscopic nonequilibrium state for systems with single-site translation symmetry. In the model under consideration, this symmetry is broken due to the Trotterization. This observation, however, raises the question, whether the observed phenomenology is unique to Trotterized dynamics or could also be seen in autonomous systems with broken single-site translation symmetry.

2- I would welcome a discussion of micromotion. Do the calculated nonequilibrium states apply only at stroboscopic times? If that is the case, how do, e.g., the profiles of the staggered magnetization shown in Fig. 1 change during one driving period?

3- The paper is clearly written and overall accessible even to non experts. However, I have found the transition from the Bethe equations (9) to the TBA in Eq. (12) rather hard to follow. As far as I understand, the Bethe equations determine the spectrum of the Floquet operator, i.e., these equations are not specific to any state of the system. However, the distribution functions appearing in Eq. (12) describe a macrostate of the system. Which state is that and how did it appear in the Bethe equations?

4- The + sign in Eq. (54) should presumably be an =.

Recommendation

Accept in alternative Journal (see Report)

---

## Round 3 · Referee Report · Anonymous (Referee 2) · 2024-12-30

Strengths

  1. the paper is clear

  2. the paper is correct

  3. integrable quantum circuits are interesting in the context of quantum computing and quantum simulations

  4. section 2 contains a useful review of TBA formulas for integrable quantum circuits

Weaknesses

  1. comparison with numerics limited to non-interacting cases

  2. rather incremental results

Report

In the past decade, many works have been dedicated to the real-time dynamics of integrable spin chains, such as the XXZ spin chain. In particular, the new theory of 'Generalized Hydrodynamics' was discovered.
Among all the works on Generalized Hydrodynamics that followed, many have focused on the so-called bipartition protocol, where two semi-infinite systems are suddenly put together, and where the predictions of Generalized Hydrodynamics are simple (because the results depend on a simple scaling variable $x/t$) and asymptotically exact.

Recently, 'integrable trotterizations' or 'integrable quantum circuits' have been introduced. These circuits are built from the 'diagonal-to-diagonal transfer matrices' known from the old algebraic Bethe Ansatz literature, with spectral parameters taken so that the transfer matrix is unitary and can be interpreted as an evolution operator. These integrable trotterizations have attracted a lot of interest recently in the context of digital quantum computation or simulation. In particular, the thermodynamic Bethe Ansatz for these systems was derived very recently ( by other people in Refs. [36-38] and by some of the authors in the Supplemental Material of Ref. [33]), as is nicely reviewed in Section 2 of this new paper.

No paper had yet looked at the bipartition protocol in integrable quantum circuits with Generalized Hydrodynamics. Now this is done, thanks to this new paper. The paper is correct, it can be published as it is.

Requested changes

Typos:
-page 4: 'Hamitlonian'
-page 5: 'masless'
-page 15: 'These points correspond to the either one of following choices'

Recommendation

Publish (easily meets expectations and criteria for this Journal; among top 50%)

---

## Round 3 · Referee Report · Anonymous (Referee 3) · 2025-3-9

Report

The Authors extend the framework of generalized hydrodynamics to integrable quantum circuits. This framework is a »classical« description of large-scale phenomena in integrable models, which possess infinitely many local conservation laws (continuity equations). It allows one to treat exactly the bipartition protocols / Riemann problems in integrable models, for example. The topic is timely due to the advent of quantum computing devices which often implement circuits similar to the ones described by the Authors herein. Although the calculations are mostly generalization of the known results in the continuous-time models, one of the results is surprising: a localized perturbation may result in global change of the system’s state, which propagates (and persists) on ballistic scales.

The Authors have assessed that two conditions for SciPost physics are met: (1) Provide a novel and synergetic link between different research areas; (2) Detail a groundbreaking theoretical/experimental/computational discovery. I am not convinced that (2) is true, but I cannot argue that (1) is not (although, as pointed out by Referee 2, the »synergetic link« has essentially been done in ref. [33]). All in all, I would lean more towards the recommendation of Referee 2, that the paper might be more suitable for SciPost Physics Core. However, since my arguments against the claim that criterion (1) is met might not be deemed really strong, I am not strongly opposed to publication in SciPost Physics either. In any case, below I provide three remarks that can slightly improve the otherwise clear and well-written paper.

Requested changes

(1) At the end of section 2.1 the Authors remark on two families of conserved quantities which break the single-site translation symmetry. They claim that the latter maps one family into the other, referring to ref. [29]. Looking at eqs. (10), (11) in the latter, their statement does not seem obvious to me. Looking at those two eqs. in the reference, for example, the densities of the first two charges are indeed translated w.r.t. each other for one site, but they have a \pm sign difference in one of the terms. Could the Authors comment on this?

(2) After eq. (59), the Authors mention a global prefactor to the eigenvalue of the propagator. I assume that prefactor is not just a phase (-1)^L, otherwise it could have been written down. Where does is come from? Looking at Faddeev’s notes, eq. (412) in arXiv:hep-th/9605187, there is no L-dependent prefactor in the definition of the quasienergy, for example.

(3) The Authors mention refs. [99-101] as examples of local perturbation evolving into a macroscopic change of the system’s state. Some other examples of such non-dispersing localized perturbations: arXiv:1207.0862, arXiv:1909.02841, arXiv:2111.06325.

Recommendation

Accept in alternative Journal (see Report)

---

## Round 4 · Author Response

We are very grateful to the Referees for their careful reading of our draft. In the following we respond to their comments.

Response to Referee 1

In the following, we present a point-by-point response to the Referee comments.

“In the past decade, many works have been dedicated to the real-time dynamics of integrable spin chains, such as the XXZ spin chain. In particular, the new theory of 'Generalized Hydrodynamics' was discovered.
Among all the works on Generalized Hydrodynamics that followed, many have focused on the so-called bipartition protocol, where two semi-infinite systems are suddenly put together, and where the predictions of Generalized Hydrodynamics are simple (because the results depend on a simple scaling variable x/t) and asymptotically exact.
Recently, 'integrable trotterizations' or 'integrable quantum circuits' have been introduced. These circuits are built from the 'diagonal-to-diagonal transfer matrices' known from the old algebraic Bethe Ansatz literature, with spectral parameters taken so that the transfer matrix is unitary and can be interpreted as an evolution operator. These integrable trotterizations have attracted a lot of interest recently in the context of digital quantum computation or simulation. In particular, the thermodynamic Bethe Ansatz for these systems was derived very recently ( by other people in Refs. [36-38] and by some of the authors in the Supplemental Material of Ref. [33]), as is nicely reviewed in Section 2 of this new paper.
No paper had yet looked at the bipartition protocol in integrable quantum circuits with Generalized Hydrodynamics. Now this is done, thanks to this new paper. The paper is correct, it can be published as it is."

We thank the Referee for their positive assessment of our work.

“Typos:
-page 4: 'Hamitlonian'
-page 5: 'masless'
-page 15: 'These points correspond to the either one of following choices'"

We thank the Referee for spotting these typos, which we have corrected in the new version of the draft.

Response to Referee 2

In the following, we present a point-by-point response to the Referee comments.

“The paper by Hübner et al. presents an interesting study of the nonequilibrium dynamics of a quantum spin chain, at the interface of three research areas that have received significant attention in recent years: (i) Trotter transitions, that is, qualitative changes in the dynamics that occur when the Trotter step size crosses a critical value; (ii) integrability and generalized hydrodynamics (GHD); (iii) quench dynamics, in particular, the bipartition protocol, which can lead to the emergence of nonequilibrium steady states. This study builds on and complements a previous work [33] that includes two of the authors of the present manuscript. The model system under investigation is an integrable quantum circuit that represents a Trotterization of the dynamics generated by the XXZ Hamiltonian. As initial states, the authors consider a single domain wall between the two halves of the spin chain in Néel, anti-Néel, or Majumdar-Ghosh states. To describe the dynamics in the space-time scaling limit, the authors build on recent developments in the theory of integrable quantum circuits, and derive the GHD equations for such systems. Apart from these technical developments, the key physical result of the paper is that for finite Trotter step sizes, the nonequilibrium macrostates appearing at late times can differ qualitatively from the continuous-time limit."

We thank the Referee for their positive assessment of our work.

"As stated above, the paper addresses a timely topic at the interface between different research areas. The results are interesting and clearly presented. I am not an expert on integrability and GHD, and, therefore, I find it hard to judge whether the paper is groundbreaking in terms of developing new techniques. However, my impression is that the foundations for the present work were laid in Ref. [33], which diminishes the claims to novelty of this work. Therefore, I believe that the paper is better suited for a less selective journal such as SciPost Physics Core."

The main criticism raised by Referee is that the novelty of our work is diminished by Ref. [33]. We do not agree with this claim, as we argue below.

First, we agree that our work does not introduce ground-breaking techniques compared to previous work on integrability. Also, it is true that Ref. [33] already highlighted the emergence of different types of behavior in integrable circuits as a function of the Trotter step. However, Ref. [33] only studied the behavior of integrable circuits in an ideal setting of translation-invariant systems, and in the late-time stationary regimes. On the contrary, our present work makes predictions on a different physical setting and studies the full time evolution (in the large space-time limit). Although one could have anticipated the emergence of peculiar behavior based on Ref. [33], it was not obvious how and to what extent it could be observed in the dynamics of simple local observables. This question is particularly relevant when considering potential benchmark implementations of integrable circuits on digital quantum devices. Our work provides a thorough study of quantum quenches in inhomogeneous settings, yielding a phenomenological overview of the possible behavior that can be observed as a function of the Trotter step. Our numerical results show very clear signatures of the Trotter transition even in the dynamics of local observables, to relatively short times.

From the technical viewpoint, we lay out the theory of GHD in integrable circuits, which was not done before. Although we do not encounter major technical complications or subtlelties in doing so, it was a priori not obvious that this would be the case. Also, our comparison of the GHD equations against independent numerics in the non-interacting regime provides convincing evidence that the GHD description applies to quantum circuits to the same degree of accuracy as previously studied continuous-time dynamics.

In summary, we believe that our work, while a natural logical continuation of the research presented in Ref. [33], is by no means a mere application of the theory developed there. Our conclusions could not have been anticipated only based on Ref. [33], and our calculations lead to results that are expected to be of interest for several communities working on non-equilibrium many-body systems, integrability, and even for future experimental work benchmarking digital quantum devices.

“1- As the authors explain, a domain wall between the Néel and anti-Néel states can be interpreted as a single localized defect, and such a defect does typically not lead to a change in the macroscopic nonequilibrium state for systems with single-site translation symmetry. In the model under consideration, this symmetry is broken due to the Trotterization. This observation, however, raises the question, whether the observed phenomenology is unique to Trotterized dynamics or could also be seen in autonomous systems with broken single-site translation symmetry."

Indeed, the Referee is right, we expect that a similar phenomenology could be observed in Hamiltonian dynamics where the Hamiltonian breaks single-site shift invariance (but preserves two-site translation symmetry). However, while this construction would appear rather ad hoc in the continuous-time limit, here it appears naturally as a consequence of increasing the Trotter step. The ability to change the global late-time behavior by a single defect in a circuit setting is also of obvious interest for benchmark experiments in current digital devices. Indeed, approximating the continuous-time evolution of a two-site shift-invariant Hamiltonian would require implementing very deep circuits, even to reach short times, which is unfeasible on current devices. Instead, existing experiments have already shown that integrable circuits can be implemented keeping sufficient coherence over a number of discrete steps that is enough to appreciate the emergence of late-time features.

We have added this discussion in the second-to-last paragraph in Sec. 4.1 (see the end of Pag. 16 and the beginning of Pag. 17 in the new version of the draft).

“2- I would welcome a discussion of micromotion. Do the calculated nonequilibrium states apply only at stroboscopic times? If that is the case, how do, e.g., the profiles of the staggered magnetization shown in Fig. 1 change during one driving period?.”

As the Referee suggests, while the quantum-circuit architecture requires that the dynamics is discrete, one could of course view the quantum-circuit dynamics as a Floquet dynamics observed at stroboscopic times. However, we are studying dynamics in the hydrodynamic limit of large time scales, namely over scales that are much larger that the single stroboscopic time step. Therefore, the calculated profiles would also exactly predict the profiles for a continuous-time evolution.

“3- The paper is clearly written and overall accessible even to non experts. However, I have found the transition from the Bethe equations (9) to the TBA in Eq. (12) rather hard to follow. As far as I understand, the Bethe equations determine the spectrum of the Floquet operator, i.e., these equations are not specific to any state of the system. However, the distribution functions appearing in Eq. (12) describe a macrostate of the system. Which state is that and how did it appear in the Bethe equations?"

This is a natural question, and we realized the text was not sufficiently clear on this point. We have added a paragraph to answer the question raised by the Referee after Eq. (18).

“4- The + sign in Eq. (54) should presumably be an =."

We thank the referee for spotting this. The + sign was correct but we forgot the =0 at the end of the equation.

---

## Round 4 · List of Changes

We have added several paragraphs to the text to address the comments of Referee 2

---

## Round 5 · Author Response

Response to Referee 3

In the following, we present a point-by-point response to the Referee comments.

“The Authors extend the framework of generalized hydrodynamics to integrable quantum circuits. This framework is a »classical« description of large-scale phenomena in integrable models, which possess infinitely many local conservation laws (continuity equations). It allows one to treat exactly the bipartition protocols / Riemann problems in integrable models, for example. The topic is timely due to the advent of quantum computing devices which often implement circuits similar to the ones described by the Authors herein. Although the calculations are mostly generalization of the known results in the continuous-time models, one of the results is surprising: a localized perturbation may result in global change of the system’s state, which propagates (and persists) on ballistic scales."

We thank the Referee for their positive assessment of our work.

“The Authors have assessed that two conditions for SciPost physics are met: (1) Provide a novel and synergetic link between different research areas; (2) Detail a groundbreaking theoretical/experimental/computational discovery. I am not convinced that (2) is true, but I cannot argue that (1) is not (although, as pointed out by Referee 2, the »synergetic link« has essentially been done in ref. [33]). All in all, I would lean more towards the recommendation of Referee 2, that the paper might be more suitable for SciPost Physics Core. However, since my arguments against the claim that criterion (1) is met might not be deemed really strong, I am not strongly opposed to publication in SciPost Physics either. In any case, below I provide three remarks that can slightly improve the otherwise clear and well-written paper."

The Referee is not convinced that our work meets the SciPost acceptance criteria. Their main criticism is the same by Referee 2, namely that this work does not introduce enough novelty compared to Ref. [33]. We strongly disagree on this point, as we explain in detail in our response to Referee 2, to which we refer. In fact, it may be useful to briefly reiterate why we believe that our work meets the SciPost acceptance criteria:
(1) it provides a link between generalized hydrodynamics (an extremely active area of theoretical and cold-atomic research over the past few years) and digital dynamics (an emerging area of research at the intersection of many-body physics and quantum-information theory) which are suitable for implementation in benchmark experiments on quantum platforms; (2) it details an important theoretical discovery: namely, a local defect in a simple discrete dynamical evolution is sufficient to cause a change in the emerging macroscopic stationary state at late times (once again, with implications on benchmark experiments on quantum devices).

“(1) At the end of section 2.1 the Authors remark on two families of conserved quantities which break the single-site translation symmetry. They claim that the latter maps one family into the other, referring to ref. [29]. Looking at eqs. (10), (11) in the latter, their statement does not seem obvious to me. Looking at those two eqs. in the reference, for example, the densities of the first two charges are indeed translated w.r.t. each other for one site, but they have a \pm sign difference in one of the terms. Could the Authors comment on this?"

We thank the Referee for spotting and pointing out this incorrect statement. What we wanted to say is that the operation leaving the set of conserved charges invariant is the combination of the single-site translation and a flip of the staggering parameter x. However, the statement as it was written in our previous version was indeed incorrect (since this statement was never used in this or the previous paper, we did not notice it). We fixed the statement in the most recent version.

"(2) After eq. (59), the Authors mention a global prefactor to the eigenvalue of the propagator. I assume that prefactor is not just a phase (-1)^L, otherwise it could have been written down. Where does is come from? Looking at Faddeev’s notes, eq. (412) in arXiv:hep-th/9605187, there is no L-dependent prefactor in the definition of the quasienergy, for example."

We thank the Referee for pointing this out. In general a prefactor may arise from the choice of normalization of the transfer matrix/evolution operator, but we have checked that in the present case such a prefactor is absent.

“(3) The Authors mention refs. [99-101] as examples of local perturbation evolving into a macroscopic change of the system’s state. Some other examples of such non-dispersing localized perturbations: arXiv:1207.0862, arXiv:1909.02841, arXiv:2111.06325."

We thank the Referee for point out these relevant references that we missed. We have added them in the most recent version of the draft.

---

## Round 5 · List of Changes

We have corrected some inaccuracies in the text, as suggested by the Referee 3

---

## Editorial Decision

published